# A Treat-to-Target Approach in IBD: Contemporary Real-World Perspectives from an International Survey

**DOI:** 10.3390/jcm14030667

**Published:** 2025-01-21

**Authors:** Mohmmed Tauseef Sharip, Biljana Brezina, Juan De La Revilla Negro, Sreedhar Subramanian, Miles Parkes, Tim Raine, Nurulamin M. Noor

**Affiliations:** 1Department of Gastroenterology, Cambridge University Hospitals NHS Foundation Trust, Cambridge CB2 0QQ, UK; mohammed.sharip@nhs.net (M.T.S.);; 2Department of Medicine, School of Clinical Medicine, University of Cambridge, Cambridge CB2 0QQ, UK

**Keywords:** monitoring, treat to target, tight control

## Abstract

**Background/Objectives**: The management of inflammatory bowel disease (IBD) varies due to differences in healthcare systems, treatment costs, access to diagnostics, and diverse clinical practices between specialists. Despite the frequent advocacy of a treat-to-target (T2T) approach, there is insufficient clarity on how clinicians implement T2T in real-world settings. We aim to conduct a large, global survey among IBD experts to identify current practices in management. **Methods**: A prospective, cross-sectional study was conducted using a 16-item survey divided into two sections—for ulcerative colitis (UC) and Crohn’s disease (CD)—and distributed to practicing IBD clinicians. **Results**: A total of 261 respondents from 88 countries participated in the survey, with the majority (253/261) being physicians and eight being IBD nurse specialists. Despite global guidance, only a quarter of the respondents routinely perform an endoscopy to assess the response after starting an advanced therapy (28.4% in UC vs. 23.5% in CD). Moreover, despite an increasing academic focus on intestinal ultrasound (IUS), 171 (66%) of respondents in UC and 132 (51%) in CD reported that they do not routinely undertake IUS to guide treatment decisions. Faecal calprotectin for monitoring treatment response was routinely used by 87% (90% in UC and 84% in CD) of the respondents. Forty-five percent reported use of therapeutic drug monitoring (TDM) both proactively and reactively and 35% reported only using TDM reactively. **Conclusions**: Our study shows considerable variation in IBD management across different countries and interpretation of the T2T approach. This highlights the need for standardised and pragmatic guidelines to help improve outcomes for patients with IBD globally.

## 1. Introduction

Crohn’s disease (CD) and ulcerative colitis (UC) are both characterised by chronic inflammation of the gastrointestinal tract. The management for both types of inflammatory bowel disease (IBD) has been revolutionised over the last two decades, with a myriad of monitoring strategies and, especially, new treatment options now available, ranging from traditional immunosuppressant medication to newer biological and small-molecule treatments [1]. Despite, and, perhaps, partly due to, this increasing array of monitoring and treatment options, management varies from country to country [2,3,4].

There are likely to be many determinants for the heterogeneity in clinical care between countries, including differences in healthcare structures, variations between local and national guidelines, and, perhaps most importantly, differences in the availability of healthcare resources. This is also on a backdrop of changing treatment focus over time. Historically, clinical remission and control of symptoms alone were the target for both patients and clinicians. Although endoscopic remission does not always correlate with clinical remission, it has been found to be a better predictor for the avoidance of disease complications compared to symptoms alone [5]. This has led to a focus on treatment algorithms with more objective targets than solely clinical remission. The STRIDE working group, in their updated STRIDE-II consensus, recommended endoscopic remission as a long-term goal (along with clinical response) to prevent future complications [6], and recent evidence has suggested that this may also be a cost effective strategy [7]. Subsequently, there has been discussion around extending this target further to include histological remission in UC and transmural remission in CD.

With this progression toward more objective measures, concepts of treat-to-target (T2T) and tight control have become more widespread in the field of IBD. A T2T approach in IBD involves directing treatment toward well-defined goals with a timely objective assessment of disease activity (“tight control”), informing tailored adjustments to treatment accordingly [8,9].

Despite recommendations on the use of T2T as part of routine IBD management [10], it is not clear how T2T approaches are being implemented around the world. We seek to gather information regarding current practice and treatment targets amongst IBD-focused clinicians from different countries by conducting a survey with a global reach.

## 2. Materials and Methods

We conducted a prospective, cross-sectional, international survey-based study at the European Crohn’s and Colitis Organisation (ECCO) annual congress, Stockholm 2024. The target population of this survey included clinicians and IBD specialist nurses looking after patients living with IBD. A 16-item electronic web-based survey was developed which contained eight questions each for both UC and CD. The survey was peer-reviewed and modified in response to feedback with subsequent endorsement by the Young ECCO (Y-ECCO), clinical research committees (ClinCom) of ECCO, and the ECCO governing board.

A hyperlink to the survey was widely circulated among attendees to the ECCO annual congress in 2024 as well as among ECCO members using the ECCO online platform, ECCO newsletter, and the ECCO electronic app. The survey was active from 17 January 2024 until 31 March 2024. Completion of the anonymous survey was deemed to imply consent for data analysis.

Only high-level baseline demographic data were collected from the respondents (country of practice, current role). The survey itself was structured into two parts, focusing respectively on UC and CD, with main sections seeking to address participants’ current experience in treating patients with IBD and the T2T approach. The complete survey questions are shown in the Appendix A.

### 2.1. Population and Inclusion Criteria

The survey sought broad input from IBD-interested clinicians and members of the multidisciplinary team around the world and represents a self-selected convenience sample.

### 2.2. Statistical Analysis

Descriptive analyses were performed on the raw dataset using python Jupyter Notebook version 6. These included responses for each categorical question. Differences among groups were assessed by chi squared test (×2) and Fisher’s exact test for dichotomous variables. All figures were produced using the following libraries: matplotlib, geopandas, and seaborn.

## 3. Results

In total, 261 participants from 88 countries fully completed the survey. Among the respondents, the majority were gastroenterologists of different grades (consultants, fellows, trainees, and resident doctors), as well as eight IBD nurse specialists. The top five countries by respondents in our survey were Italy, United Kingdom (UK), India, Spain, and Germany (Figure 1).

### 3.1. Objective Evaluation of Inflammatory Disease

In the context of UC, the majority of the respondents (225/261 [86.2%]), reported performing either a colonoscopy or flexible sigmoidoscopy before escalating to an advanced therapy. Only a small proportion of the respondents (36/261 [13.8%]) escalated treatment to advanced therapy based on reports of symptoms from patients alone (Figure 2A). In the context of CD, methods for objective evaluation were more diverse depending on disease location, i.e., isolated small bowel or upper GI disease vs. colonic disease; however, colonoscopy was still the preferred mode of evaluation (175/261 [67.3%]), followed by MRI enterography (MRE) in 59/261 (22.7%) and abdominal CT scan (15/261 [5.8%]) (Figure 2B). Around half of the respondents (164/261 [62.8%] in UC, 136/261 [52.3%] in CD) assessed the histology prior to treatment escalation decisions (Appendix A).

### 3.2. Role of Intestinal Ultrasound

A minority of the respondents (90/261 [34.5%]) reported that they routinely perform intestinal ultrasound (IUS) to dictate treatment in UC patients (Figure 3A). This number was numerically higher for CD (109/261 [41.9%]; *p* = 0.104), as shown in Figure 3B.

When evaluating the top five countries based on clinical practice of the respondents regarding use of IUS in the context of both UC and CD, wide variations in practice were noted among countries (Appendix A). The use of IUS was less commonly reported in the context of UC, overall, with no respondents reporting availability in the UK. The Fisher’s exact test showed the odds of performing an IUS in the context of both UC and CD was significantly lower in the UK and India, whereas the association was positive in Italy and strongest in Germany (Appendix A). A chi-square test also indicated a significant association between the use of IUS and the country of origin, *p* < 0.001 for both UC and CD (Appendix A).

### 3.3. Post-Treatment Monitoring and Assessment

In terms of monitoring for treatment response in IBD (Figure 4), only a quarter of the respondents routinely perform an endoscopy after starting an advanced therapy (28.4% in UC vs. 23.5% in CD). In contrast, the use of patient-reported outcomes (PROs) was widely reported (249/261 [95.4%] in UC, (240/261 [91.9%] in CD). Commonly used PROs included the ones collecting information about symptoms, such as PRO2, PRO3, patient-reported Harvey–Bradshaw index (HBI) score for CD and patient-reported simple clinical colitis activity index (SCCAI) for UC. PRO symptom targets were widely used and followed closely by faecal calprotectin (235/261 [90%] in UC, 219/261 [83.9%] in CD) and CRP (209/261 [80%] in UC, 217/261 [83.1%] in CD).

### 3.4. Therapeutic Drug Monitoring

With regard to TDM for anti-tumour necrosis factor (TNF) treatment, 44.8% (117/261) reported checking TDM both proactively and reactively in the context of UC, and 36.4% (95/261) reported performing TDM only reactively. The pattern was similar for CD (Appendix A). When evaluating the pattern of TDM performance in the five top countries by responders, respondents from the UK more commonly reported routinely performing TDM both proactively and reactively, whereas respondents from Italy, India, and Germany typically only performed TDM reactively (Appendix A). The association between the type of TDM performed and the country of origin was significant (χ^2^ statistic: 14.2, *p* = 0.0067 for UC, χ^2^ statistic:16.4, *p* = 0.002 for CD; Appendix A). The Fisher’s exact test showed that the OR of performing TDM proactively and reactively in the UK was 4.75 (*p* = 0.005 in UC; *p* = 0.004 in CD), whereas the OR in India to perform TDM proactively and reactively was 0.0 (*p* < 0.005 in UC; *p* = 0.001 in CD; Appendix A).

Treatment targets after the first year of starting advanced therapy were then considered (Figure 5). The commonest response was to indicate that the treatment aim after one year of starting advanced therapy for UC is endoscopic remission (92/261, 35%), as shown in Figure 5A. Only 30/261 respondents (11.5%) indicated that steroid-free remission is their treatment goal. Notably, among participants from the top five responding countries, there were variations in terms of treatment target after the first year of starting advanced therapy (Figure 5B). For CD, 75/261 respondents (28.7%) aimed for endoscopic remission (Figure 5C). Additionally, 46/261 participants (17.6%) targeted transmural remission using MRI, CT, or IUS modalities, and 23/261 participants (8.8%) focused solely on PROs as treatment outcomes. Once again, there were variations in terms of treatment targets among the top five countries of the respondents (Figure 5D).

The majority of the IBD practitioners reported that they do not routinely stop advanced therapies (Figure 6). A total of 127 out of 261 respondents (48.6%) indicated that they do not consider elective discontinuation of advanced therapy and they simply continue the advanced therapy. In comparison, 109 respondents (41.7%) stated that they do discuss with their patient about stopping treatments, but their personal preference is for treatment continuation. This pattern was consistent for both UC and CD. Only 24 respondents (9.1%) reported that they would routinely consider stopping advanced therapy after achieving remission.

## 4. Discussion

This survey of IBD clinicians is one of the largest global surveys to date examining contemporary real-world practices for treatment targets used in IBD management. Our survey findings highlight considerable heterogeneity in the practice for treatment escalation decisions and monitoring practices. The findings demonstrate that the majority of IBD specialists do perform some form of objective evaluation of IBD disease activity after treatment escalation, aligning with the treat-to-target approach outlined by the STRIDE II consensus group [9].

Although widely advocated, there have been few studies examining real-world implementation of a treat-to-target approach to IBD management around the world. A previous single-country survey on the management of 246 patients with UC has shown that there is limited uptake of “treat-to-target” recommendations [11]. After initiation of the treatment, a global survey of 359 participants from 60 countries also demonstrated limited real-world use of either endoscopy or histology to assess the response to treatment in the context of UC [12]. Indeed, a survey of IBD specialists from 40 countries around Europe highlighted the impact of resources on the ability to monitor IBD activity, with greater resources for monitoring being afforded in high-income countries [13]. A further international survey of 195 respondents from 38 countries also highlighted the major impact of resource availability and how low access to investigations impacted on the ability to deliver a treat-to-target approach [14]. Importantly, none of these previous surveys specifically addresses all of the wider topics around treatment escalation, details regarding monitoring practices, or the potential for treatment de-escalation.

Control of disease activity has consistently been shown to be associated with better outcomes for patients with both UC and CD [15]. While symptoms and signs may be indicative of ongoing disease activity in the context of both UC and CD, it is now widely recognised that there can be a disconnect between symptoms and disease activity. Recent evidence has suggested that approximately 50% of patients have suboptimal disease control and that many patients with ongoing active inflammation can be asymptomatic [16]. The discordance between the symptoms and the mucosal inflammation is more significant in CD patients compared to UC [17]. Given the poor correlation between symptoms and inflammatory activity, it is usually advised that treatment escalation be corroborated by objective evidence of intestinal inflammation [18]. Although widely recommended, there have been few studies specifically assessing the role and frequency of endoscopic evaluation before starting an advanced therapy, and even fewer studies examining the longer-term benefit from routine endoscopic monitoring in terms of likelihood of remission or prevention of complications [19].

In our survey, 86.2% of the respondents reported routinely performing either a colonoscopy or flexible sigmoidoscopy before escalating to an advanced therapy in the context of UC. The choice of evaluation prior to escalation in the context of CD was more variable; this may be because of greater variations in disease location, including areas difficult to reach by endoscopic investigation. However, even with respect to CD, the most commonly used method of evaluation prior to treatment escalation is ileo-colonoscopy, followed by cross-sectional imaging with either MRI or CT. It is clear from our survey that distinct tools appear to be used prior to treatment escalation and compared to those for treatment monitoring.

The STRIDE II consensus recommends a combination of PROs, biomarkers, endoscopy, and imaging to monitor the treatment response in IBD patients and the use of a tight control approach [9]. Indeed, the CALM trial previously demonstrated that tight control using a combination of clinical symptoms with inflammatory biomarkers resulted in better clinical and endoscopic outcomes than a symptom-driven approach alone [20]. However, it is important to note that CALM was performed in a cohort of patients with moderate to severe CD, so whether such findings would also be applicable to UC and milder or asymptomatic CD is unknown [21]. When respondents were asked about optimal methods of follow-up in our survey, the majority did report using a combination of PROs and objective markers, with faecal calprotectin reported most commonly. This would be in line with a previous survey-based study highlighting that faecal calprotectin is the most commonly available and used biomarker for inflammatory disease globally, although there was some variation among countries [14]. However, somewhat surprisingly, 31% of survey respondents reported that they were relying on symptoms alone. This may be related to the variations in acceptability of currently available monitoring tools. A previous nationwide survey study showed that lower acceptability of the faecal calprotectin was mostly due to embarrassment for collection or transport of the stool [22]. The same study also showed that acceptance of colonoscopy/sigmoidoscopy was the lowest as an objective assessment tool, whereas cross-sectional imaging with IUS and MRE performed significantly better in comparison. Therefore, the mode of objective assessment may not only be dependent on the resources of the healthcare setting and the training available, but also on acceptability among the patients.

Most contemporary international guidelines recommend an endoscopic investigation to assess the response to treatment, typically recommended after the induction doses of biologics or small molecules [23,24,25,26,27]. However, in line with other studies examining real-world clinical practice, we demonstrate that the majority of IBD clinicians did not report routinely performing an endoscopy to assess the response to induction with a medical treatment [28]. This disconnect between clinical practice and global guidelines is important to understand. This may reflect the fact that both clinicians and patients may not feel the need to undertake an endoscopic procedure if a patient is asymptomatic and has otherwise normal objective markers of inflammation, such as a faecal calprotectin. A further reason for the relatively low uptake of the endoscopic assessment may be the invasive nature and need for bowel preparation of a full colonoscopy—both factors impair acceptability of colonoscopy. It is also worth highlighting that the accessibility of the endoscopy is challenging in many healthcare settings, e.g., in the UK, there has been a large backlog of endoscopy post pandemic, with many patients waiting months or even years for their procedure to take place if referred for a non-cancer indication [29]. Although a minority, it is important to note that there was still a sizeable number of respondents who did perform endoscopic evaluation after the induction treatment with an advanced therapy (28.4% in UC and 23.5% in CD).

There has been rising interest in the use of IUS to make treatment escalation decisions and for monitoring treatment response. A previous study has shown that IUS, MRE, and CT enterography have a similar diagnostic accuracy for CD complications such as strictures, abscesses, and fistulas [30]. A cohort study showed that a combination of faecal calprotectin and IUS could accurately predict histological disease activity in the context of UC [31]. There are multiple factors to support the use of IUS over other investigations, including its non-invasive nature, its relatively low cost, its accuracy to predict intestinal inflammation, particularly of the small bowel [32], and its ability to perform as a point of care investigation. Despite the increasing academic focus on IUS, this is not routinely performed nor widely available for many respondents in our survey, although there was geographic variation, likely reflecting available resources and facilities.

The concept of disease clearance is relatively novel, particularly in the context of UC [16]. Disease clearance is defined as a combination of clinical remission, endoscopic remission, and histological remission [33]. Despite the increasing interest, the most recent STRIDE II consensus does not recommend histological remission as a formal treatment target in the context of UC [9]. Although there is some growing evidence that histological remission may offer additional benefits over endoscopic remission in predicting long-term remission and cancer prevention in the context of UC [34], aiming for this target may come at a cost of increased immunosuppression and potentially higher costs from cycling between medications more rapidly.

When the respondents were asked if they routinely check histology before treatment escalation, the majority responded that they do (164/261 [62.8%] in UC and 136/261 [52.3%] in CD). However, the use of histology as a tool to monitor response was far less common (86/261 [32.1%] in UC, 53/261 [20.3%] in CD). The ongoing VERDICT trial should help answer the question of whether treating the target of histological remission in the context of UC is beneficial or not [35]. It is also not currently clear whether treating a histological target in the absence of symptoms is beneficial for all patients or not.

It is clear that future monitoring studies will need to compare the impact on long-term outcomes of the use of different modalities and consider the relative cost and resource availability in the context of the T2T strategy. An economic analysis of the CALM study showed that tight control management of CD is associated with fewer hospitalisations and quality-adjusted life years gain, resulting in good value for money [36]. However, that analysis was performed based on tight control of symptoms and biomarkers alone in the healthcare settings of the UK. Its applicability in the broader context of a T2T strategy, which involves a more objective assessment, i.e., colonoscopy/flexible sigmoidoscopy, MRE, CT, and IUS, in different health care settings needs further evaluation. Equally, the ongoing debate on switching an advanced therapy in an asymptomatic patient with isolated high biomarkers may contribute to the reluctance of some clinicians to adopt the T2T strategy [21].

The seminal PANTS observational study demonstrated a dose–response relationship between week 14 anti TNF levels and treatment failure across 3 years of treatment in CD patients [37]. Despite a large body of observational data supporting the use of TDM after initiation of anti-TNF medications, neither reactive nor proactive TDM was consistently utilised, with only 44% of respondents performing any form of TDM. Interestingly, the majority of the respondents from the UK reported performing both proactive and reactive TDM, whereas, in other countries, the use of proactive TDM was more limited. The limitation of performing TDM is likely due to a combination of several factors, which have been identified from a previous cross-sectional study among rheumatologists, including: (i) observing clinical needs; (ii) understanding how testing can improve practice; (iii) insufficient clinical evidence; (iv) insufficient resources to pay for testing; and (v) insufficient capability to deliver testing [38].

The final aspect of our survey sought to understand the real-world practice of treatment de-escalation in patients where remission has been achieved. Interestingly, and perhaps as a result of the data from SPARE, STOP-IT, and HAYABUSA, all of which showed a high rate of relapse (up to 50%) after stopping the anti-TNF therapy [39,40,41], the majority of the respondents reported that they do not routinely consider stopping advanced therapy (48.8% in UC, 52.3% in CD).

It should be noted that, in general, there were limited numbers of respondents from each country and from outside Europe. Caution should, therefore, be applied before attempting to generalise these results to other countries or across continents. Also, due to an unknown sampling frame, it is not possible to know how many individuals responded out of the number invited, as is often the case with large global surveys, particularly when disseminated by an international organisation. The survey was also offered only in English, and translation into multiple languages could have, potentially, produced a wider global input. This was a survey-based study and, as it is common among all survey-based studies, there is a high possibility of bias. This includes risk for both response bias, where participants may provide socially desirable answers, and selection bias, given that the majority of the respondents were IBD specialists and mostly doctors, with only eight nurse respondents. It is also not clear if practice might be even more variable among non-IBD specialist doctors. A timeline of when clinicians use objective measures to monitor disease activity, although clinically interesting, was not specifically asked about in this study.

Despite the above, this cross-sectional survey does have strengths, including it being, to our knowledge, the largest survey of real-world treat-to-target practice of its kind in the field of IBD, with respondents from 88 countries around the world, and providing novel data exploring clinician decision-making around escalation, monitoring, and de-escalation. An important aspect of future work could be to better understand how clinicians are utilising advanced therapies within a T2T approach, and what specific challenges are faced by clinicians in different countries.

## 5. Conclusions

In conclusion, we found highly heterogenous approaches for decision-making among countries for IBD management. In line with the STRIDE-II consensus guidelines, the majority of IBD clinicians indicated that they use a combination of objective evaluations in addition to symptoms to help with treatment escalation decisions; however, the actual objective measures vary and likely depend on local practices as well as resource availability. Only a quarter of the respondents routinely perform an endoscopy to assess response after starting an advanced therapy, highlighting the disconnect between clinical practice and many global guidelines.

Our survey provides a large real-world illustration of differences in contemporary practices around the use of treatment targets for treatment escalation decisions and for the monitoring of treatment response. It is clear that, going forward, any recommendations for future treatment targets need to be realistic and achievable across a range of healthcare settings as well as being patient centred.

## Figures and Tables

**Figure 1 jcm-14-00667-f001:**
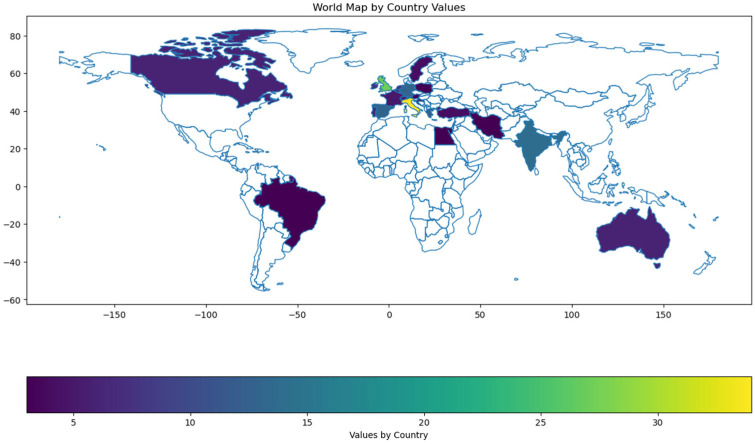
Respondents by countries.

**Figure 2 jcm-14-00667-f002:**
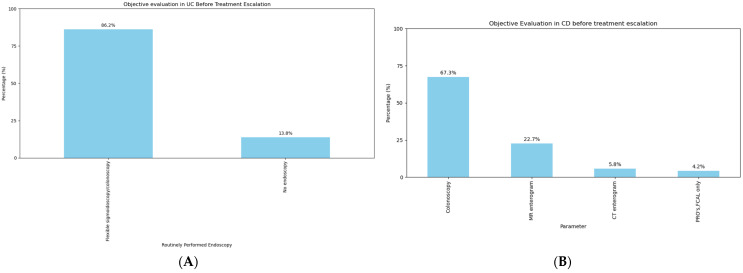
Objective evaluation of IBD before starting an advanced therapy in the context of (**A**) ulcerative colitis and (**B**) Crohn’s disease.

**Figure 3 jcm-14-00667-f003:**
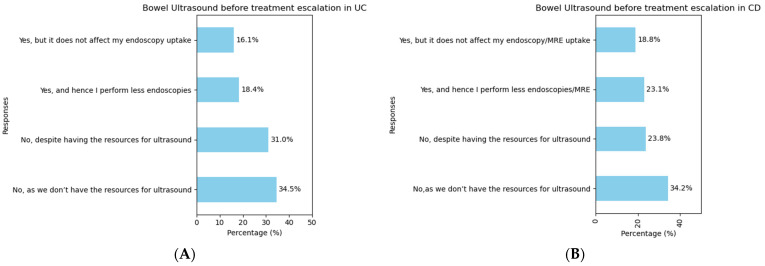
Use of intestinal ultrasound in the context of (**A**) ulcerative colitis and (**B**) Crohn’s disease.

**Figure 4 jcm-14-00667-f004:**
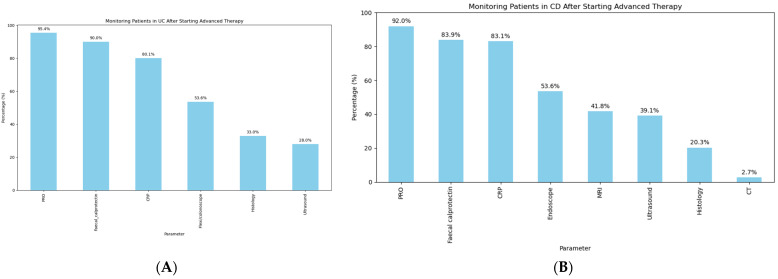
Monitoring of patients after starting an advanced therapy in the context of (**A**) ulcerative colitis and (**B**) Crohn’s disease.

**Figure 5 jcm-14-00667-f005:**
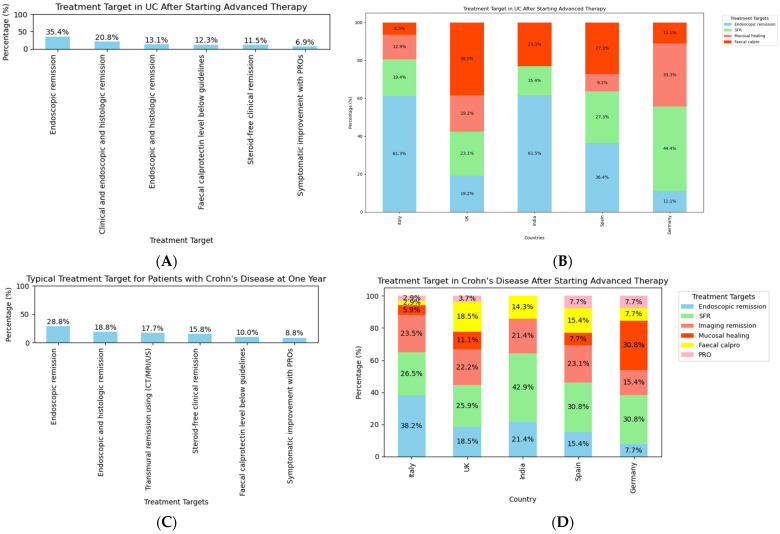
Treatment target after one year of starting advanced therapy in the context of (**A**) ulcerative colitis, (**B**) based on the top five countries of the respondents, and in the context of (**C**) Crohn’s disease, (**D**) based on the top five countries of the respondents.

**Figure 6 jcm-14-00667-f006:**
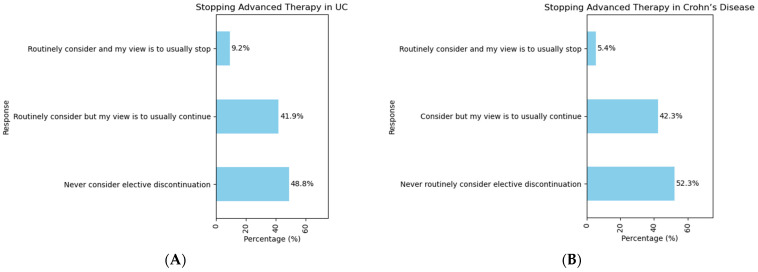
Clinical practice regarding the interruption of advanced therapy in the context of (**A**) ulcerative colitis and (**B**) Crohn’s disease.

## Data Availability

The data underlying this article are available upon reasonable request.

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
