# Peer review of "A Treat-to-Target Approach in IBD: Contemporary Real-World Perspectives from an International Survey"

_jcm, 2025, doi:10.3390/jcm14030667_

Round 1
Reviewer 1 Report
Comments and Suggestions for Authors
Very comprehensive and interesting paper. Would be interesting to have a timeline showing when IBD clinicians start to monitor their patients with IUS, enterography and endoscopy.
Author Response
|
Reviewer 1: Very comprehensive and interesting paper. Would be interesting to have a timeline showing when IBD clinicians start to monitor their patients with IUS, enterography and endoscopy. |
|
Response 1: We would like to thank reviewer 1 for taking time to review our paper and for the kind comments. We agree with the reviewer that having a timeline in terms of performing IUS, enterography and endoscopy would have been interesting. When designing the cross-sectional survey and in the interests of ensuring brevity, we did not specifically ask about timing of investigations. We have now added the following sentence to the limitations section of our article –
“Most IBD clinicians in this study performed an objective evaluation after starting an advanced therapy. Although our study provides an indication of which investigations are used, we did not specifically ask about timeline of when these investigations are used.“
|

Reviewer 2 Report
Comments and Suggestions for Authors​IBD management across different countries is determined by the capabilities of the health service. If it is not known what it is about, it is about money and thus the way of financing the treatment of citizens by the country's governments. We can have excellent guidance, clear definition of T2T strategy but... Countries are ruled by politicians and politicians are not doctors. Publications of this type are necessary to make decision-makers aware of the realities and standards in the surrounding reality.
Great idea and congratulations on the article
Comments on the Quality of English LanguageHigh standard of Eng language...
Author Response
Reviewer 2: ​IBD management across different countries is determined by the capabilities of the health service. If it is not known what it is about, it is about money and thus the way of financing the treatment of citizens by the country's governments. We can have excellent guidance, clear definition of T2T strategy but... Countries are ruled by politicians and politicians are not doctors. Publications of this type are necessary to make decision-makers aware of the realities and standards in the surrounding reality.
Great idea and congratulations on the article
Response 2: We would like to thank reviewer 2 for their detailed review and kind comments on our study. We entirely agree with the comments and hope that this publication may help to support improvements and harmonization of quality of care standards for patients living with IBD across different countries.
Reviewer 3 Report
Comments and Suggestions for Authors
This study evaluates global practices in implementing the treat-to-target (T2T) approach for managing inflammatory bowel disease (IBD), focusing on variations across healthcare systems and clinical settings. Through a survey of 261 IBD specialists from 88 countries, the authors explore how diagnostic tools, endoscopy, intestinal ultrasound, and therapeutic drug monitoring are utilized. The findings reveal considerable inconsistencies in applying T2T principles, highlighting the need for standardized guidelines to improve IBD care.
This is an interesting study, shedding light on the disparities in the adoption of T2T principles across diverse clinical practices. The STRIDE guidelines represent a significant advancement in IBD care, and exploring their practical implementation is a timely and valuable endeavor. However, some critical areas need further exploration and discussion to enhance the study’s relevance and impact.
It is surprising that so few physicians routinely use colonoscopy to assess disease activity, given its central role in IBD management. Do the authors have any insight into why this practice is not more widespread? It would also be helpful to know the circumstances under which endoscopy is typically performed and whether barriers such as resource availability or patient compliance are factors.
The paper lacks sufficient discussion on how physicians approach treatment with the numerous advanced therapies available. How comfortable are they with integrating therapies like biologics or small molecules into a T2T framework? Understanding this would provide a more comprehensive picture of real-world practices.
While the survey highlights variability in monitoring tools such as intestinal ultrasound and therapeutic drug monitoring, it would be beneficial to discuss the practical implications of these findings. For example, are some tools underutilized due to limited access, or are they considered less critical in certain clinical settings? Addressing these aspects would strengthen the conclusions.
- Expand on Endoscopy Insights: Investigate why so few physicians use colonoscopy for treatment monitoring and provide specific data on when they typically perform it.
- Address Treatment Practices: Discuss how clinicians are utilizing advanced therapies in the T2T approach and what challenges they face.
- Broaden Practical Implications: Analyze the barriers to standardizing tools like intestinal ultrasound and TDM in real-world settings.
Author Response
Reviewer 3: Comment 1: This study evaluates global practices in implementing the treat-to-target (T2T) approach for managing inflammatory bowel disease (IBD), focusing on variations across healthcare systems and clinical settings. Through a survey of 261 IBD specialists from 88 countries, the authors explore how diagnostic tools, endoscopy, intestinal ultrasound, and therapeutic drug monitoring are utilized. The findings reveal considerable inconsistencies in applying T2T principles, highlighting the need for standardized guidelines to improve IBD care. This is an interesting study, shedding light on the disparities in the adoption of T2T principles across diverse clinical practices.
Response: We would like to thank reviewer 3 for their comments and recognising the interest that our article will likely generate among colleagues in the field and how our study helps to shed light on disparities in the adoption of T2T principles.
The STRIDE guidelines represent a significant advancement in IBD care, and exploring their practical implementation is a timely and valuable endeavor. However, some critical areas need further exploration and discussion to enhance the study’s relevance and impact. It is surprising that so few physicians routinely use colonoscopy to assess disease activity, given its central role in IBD management. Do the authors have any insight into why this practice is not more widespread? It would also be helpful to know the circumstances under which endoscopy is typically performed and whether barriers such as resource availability or patient compliance are factors.
- Expand on Endoscopy Insights: Investigate why so few physicians use colonoscopy for treatment monitoring and provide specific data on when they typically perform it.
Response: We thank the review for their comments. Similarly, we were also surprised to see only a small number of clinicians reporting that they routinely perform colonoscopy for treatment monitoring. Given that this was a cross-sectional survey study, and the anonymous nature of completion, unfortunately we are unable to go back to the same sampling frame to identify the root cause of why fewer endoscopies are being performed. We have speculated in the article, and based on the clinical experience of all co-authors, that this may at least in part be due to the widespread availability and generally acceptable sensitivity of non-invasive objective measures such as faecal calprotectin. To cover the above we have now added and modified the following text to the manuscript:
“This may reflect the fact that both clinicians and patients may not feel the need to undertake an endoscopic procedure if a patient is asymptomatic and has otherwise normal objective markers of inflammation, such as a faecal calprotectin. A further reason for the relatively low uptake of endoscopic assessment may be the invasive nature and need for bowel preparation of a full colonoscopy – both factors which impair acceptability of colonoscopy. It is also worth highlighting that the accessibility of the endoscopy is challenging in many healthcare settings, e.g., in the UK, there has been a large backlog of endoscopy post-pandemic, with many patients waiting months or even years for their procedure to take place if referred for a non-cancer indication.”
We also added the following limitation in our study:
“Most IBD clinicians in this study performed an objective evaluation after starting an advanced therapy. Although our study provides an indication of which investigations are used, we did not specifically ask about timeline of when these investigations are used.“
Comment 2: Address Treatment Practices: Discuss how clinicians are utilizing advanced therapies in the T2T approach and what challenges they face.
Response: We thank the review for this comment and agree it is an interesting topic. As this study reports on a cross sectional survey, unfortunately we are unable to go back to the same sampling frame to specifically ask about this. However we have now added the following to the discussion section, outlining important future work that could be conducted in this area.
“An important aspect of future work could be to better understand how clinicians are utilizing advanced therapies within a T2T approach, and what specific challenges are faced by clinicians in different countries.“
Comment 3: Broaden Practical Implications: Analyze the barriers to standardizing tools like intestinal ultrasound and TDM in real-world settings.
Response: We thank the reviewer for this important point. We have now added the following in sections as part of the discussion within the article.
TDM
“The limitation of performing TDM is likely due to combination of several factors, which have been identified from a previous cross-sectional study among rheumatologists, including: (i) observing clinical need; (ii) understanding how testing can improve practice; (iii) insufficient clinical evidence; (iv) insufficient resources to pay for testing; and (v) insufficient capability to deliver testing.”
IUS
“Despite the increasing academic focus on IUS, this was not routinely performed nor widely available for many respondents in our survey, although there was geographic variation, likely reflecting available resources and facilities.”
Reviewer 4 Report
Comments and Suggestions for Authors
Comments to Author,
1. The author presented the data in confusing way and try to use some standard software to make graph and significance between the groups.
2. The author written well but in results section, I feel like some kind of conversion and its not like scientific manuscript.
3. In conclusion, try to provide the clear-cut conclusion, the current conclusion is confusing.
4. Provide the table with incidence of IBD in varies Conterie's.
Author Response
Reviewer 4. The author presented the data in confusing way and try to use some standard software to make graph and significance between the groups.
Response: We thank the reviewer for their time to review our manuscript. We have clarified in the methods and made clearer that we used Python for analysis and the graphs were created using a standard library. We have outlined the analyses used and the scientific rationale for using them.
Comment 1: The author written well but in results section, I feel like some kind of conversion and its not like scientific manuscript.
Response: We thank the reviewer for this comment and for recognising the manuscript is well-written. We also highlight the positive comments from the above reviewers. Reviewer 1: Very comprehensive and interesting paper. Reviewer 2: Publications of this type are necessary to make decision-makers aware of the realities and standards in the surrounding reality. Reviewer 3: The findings reveal considerable inconsistencies in applying T2T principles, highlighting the need for standardized guidelines to improve IBD care. This is an interesting study. We note the broadly positive comments with regards clarity of findings and the manuscript from across other reviewers.
Comment 2: In conclusion, try to provide the clear-cut conclusion, the current conclusion is confusing.
Response: We thank the reviewer for this comment and agree it is helpful to have an important take-home message from the paper. We have now accordingly modified the conclusion to make clear the main message:
“Our survey provides a large real-world illustration of differences in contemporary practices around the use of treatment targets for treatment escalation decisions and for monitoring of treatment response.”
Comment 3: Provide the table with incidence of IBD in varies Conterie's.
Response: Spelling mistakes are noted, and we believe the reviewer may have been suggesting to report incidence of IBD in various countries rather than “varies Conterie’s”. We thank the reviewer for their suggestion but believe this is beyond the remit of the article. Moreover, this cross-sectional survey had respondents from 88 countries and we do not believe it will be either helpful for readers or add anything to the interpretation of findings from this article to add lengthy text descriptors of incidence rates across 88 countries around the world.
|
Response to Comments on the Quality of English Language |
|
|
|
Response 1: We note that reviewer 1,2,3 all reported no issues with English language. The only selection for issues about English language reporting was from reviewer 4, but we also note comments from reviewer 4 that this was a well-written paper, so believe the reviewer may have selected this option in error. |